# Development and evaluation of a novel music-based therapeutic device for upper extremity movement training: A pre-clinical, single-arm trial

Nina Schaffert[1,2]*, Thenille Braun Janzen[3], Roy Ploigt[2], Sebastian Schlüter[2], Veronica Vuong[4], Michael H. Thaut[4]

1 Department of Movement and Training Science, Institute for Human Movement Science, University of Hamburg, Hamburg, Germany, 2 BeSB GmbH Berlin, Sound Engineering, Berlin, Germany, 3 Center for Mathematics, Computing and Cognition, Universidade Federal do ABC, São Bernardo do Campo, Brazil, 4 Music and Health Science Research Collaboratory, Faculty of Music, University of Toronto, Toronto, Canada

* nina.schaffert@uni-hamburg.de

**Data Availability Statement:** All relevant data are within the paper and its Supporting Information files.

## Abstract

Restoration of upper limb motor function and patient functional independence are crucial treatment targets in neurological rehabilitation. Growing evidence indicates that music-based intervention is a promising therapeutic approach for the restoration of upper extremity functional abilities in neurologic conditions such as cerebral palsy, stroke, and Parkinson's Disease. In this context, music technology may be particularly useful to increase the availability and accessibility of music-based therapy and assist therapists in the implementation and assessment of targeted therapeutic goals. In the present study, we conducted a pre-clinical, single-arm trial to evaluate a novel music-based therapeutic device (SONATA) for upper limb extremity movement training. The device consists of a graphical user interface generated by a single-board computer displayed on a 32" touchscreen with built-in speakers controlled wirelessly by a computer tablet. The system includes two operational modes that allow users to play musical melodies on a virtual keyboard or draw figures/shapes whereby every action input results in controllable sensory feedback. Four motor tasks involving hand/finger movement were performed with 21 healthy individuals (13 males, aged 26.4 ± 3.5 years) to evaluate the device's operational modes and main features. The results of the functional tests suggest that the device is a reliable system to present pre-defined sequences of audiovisual stimuli and shapes and to record response and movement data. This preliminary study also suggests that the device is feasible and adequate for use with healthy individuals. These findings open new avenues for future clinical research to further investigate the feasibility and usability of the SONATA as a tool for upper extremity motor function training in neurological rehabilitation. Directions for future clinical research are discussed.

**Funding:** The authors received no specific funding for this work. RP and SS are affiliated to a commercial company: BeSB GmbH. The funder provided support in the form of salaries for authors RP and SS but did not have any additional role in the study design, data collection and analysis, decision to publish, or preparation of the manuscript. The specific roles of these authors are articulated in the 'author contributions' section.

**Competing interests:** TBJ and VV declare no competing interests. RP and SS are employed engineers at BeSB GmbH Berlin, and NS currently serves as unpaid consultant and informal scientific advisor for BeSB GmbH Berlin. The device presented and tested in this study is a product developed in collaboration between BeSB and MT with potential for commercialization. This commercial affiliation does not alter our adherence to PLOS ONE policies on sharing data and materials.

## Introduction

Effective use of the arm and hand to reach, grasp, release, and manipulate objects is often compromised in individuals with neurologic disorders such as cerebral palsy [1], stroke [2, 3], Parkinson's Disease [4, 5], among others. Impairments of upper extremity function include reduced muscle power, sensory loss, increased muscle spasticity, and lack of motor control [1, 6–8], resulting in significant long-term functional deficits with relevant impact on patients' activities of daily living, independence, and quality of life [9–12]. Therefore, improving upper limb functional abilities and promoting functional independence are crucial treatment targets for neurological rehabilitation.

Functional restoration of the upper extremity is thought to be achieved through a combination of neurophysiological and learning-dependent processes that involve targeted training to restore, substitute, and compensate the weakened functions [13, 14]. Frequently reported neurorehabilitation approaches for upper limb movement in cerebral palsy [15, 16], stroke [13, 17], and Parkinson's Disease [18, 19] include standard treatment methods such as general physiotherapy (i.e., muscle strengthening and stretching), constraint-induced movement therapy and bimanual training, as well as technology-based approaches (i.e., virtual reality, games, and robot-assisted training) [20–26] and music-based interventions [27–29].

There is growing evidence that music-based interventions are a promising therapeutic approach for the restoration of upper extremity functional abilities in neurologic conditions including stroke [30, 31], cerebral palsy [32], and Parkinson's Disease [28, 33]. For instance, there is extensive research on the effectiveness of therapeutic techniques such as Music-supported Therapy and Therapeutic Instrumental Music Performance in rehabilitating arm paresis after stroke through musical instrument playing [30, 34–40]. Similarly, active musical instrument playing (i.e., piano) also seems to improve manual dexterity and finger and hand motor function in individuals with cerebral palsy [32, 41–43]. Furthermore, consistent evidence indicates that interventions using rhythmic auditory cues or rhythmically-enhanced music are effective to increase muscle activation symmetry [44], improve range of motion and isometric strength [45], enhance spatiotemporal motor control [46], and decrease compensatory reaching movements [44].

Music-based movement rehabilitation for upper limb training is particularly interesting because playing a musical instrument provides real-time multisensory information that enhances online motor error-correction mechanisms and supplements possible perceptual deficits [47–49]. Research has also shown that the engagement of multisensory and motor networks during active music playing promotes neuroplastic changes in functional networks and structural components of the brain, which are crucial neurophysiological processes for neurologic recovery [50–53]. In addition, there is robust evidence that the use of metronome or beat-enhanced music is important to support movement training as the continuous-time reference provided by the rhythmic cues allow for movement anticipation and motor preparation, bypassing the movement timing dysfunction through the activation of alternate or spared neural pathways [33, 54]. Finally, emotional-motivational aspects of music-making also play a significant role in the rehabilitating effects of music-based intervention through music-induced changes in mood, arousal, and motivation [27, 55], with potential effects on perceived physical endurance and fatigue [30, 56].

Traditionally, music-based interventions for the rehabilitation of upper extremity generally involve the use of acoustic musical instruments such as guitar, piano, and pitched and non-pitched percussive instruments [38, 42]. However, traditional instruments can impose limitations for those with significant cognitive or physical impairments as they require more resistance to press a key or to move a string and are less adaptable to the patient's needs. Recently,

studies have acknowledged the relevance of music technology to increase the availability and accessibility of music-based therapy for patients with neurological disorders in different settings, including hospitals, communities, and home environment [57–60]. For example, the use of programmable devices can help patients to exercise independently in addition to scheduled caregiver-guided sessions, thus increasing treatment intensity [58]. Digital music and sound devices can provide enhanced auditory feedback to kinematic movement components such as velocity and acceleration, range of motion, joint angles, spatial and temporal limb trajectories, even in stages of limited physical movement capability [57]. Additionally, technology may assist therapists in the implementation of individual therapeutic goals and provide immediate assessment of measurable changes with objective outcome measures (e.g., total movement time, movement variability, force, inter-response interval).

The introduction of music technology with the use of digital musical instruments, such as keyboards and drum pads [30, 38] and, more recently, touchscreen devices (e.g. tablets) using commercially available music software [40, 61, 62], have provided novel approaches for the application of active musical instrument playing in the rehabilitation of upper extremity motor function. For instance, electronic keyboards and digital sound surfaces enable users with complex needs the possibility to play a musical instrument in an adapted form to train fine and gross movements of the paretic extremity [30, 38]. However, the therapeutic sessions are commonly provided by a therapist at a rehabilitation center or hospital, thus limiting its availability for additional and independent at-home-practice. The use of mobile tablets in music therapy has notable advantages in this regard, as they provide affordable, accessible, and portable alternatives to digital music instruments. However, there is a lack of hardware and software developed specifically for clinical practice, and the use of touchscreen devices in music therapy is often limited by resources developed for the wider consumer market [63]. Therefore, there is a clear need for the development of new technology to address this important gap in music-based neurologic rehabilitation. In light of this need, a novel music-based therapeutic device for upper extremity movement training was developed with the ultimate goal to improve upper extremity motor function, to increase independent patient engagement, to enhance treatment quality, intensity, and compliance, and to assist therapists during treatment implementation and assessment.

The objective of this study is to describe a novel music-based therapeutic device called SONATA and to conduct a pre-clinical, single-arm trial to test the device with healthy individuals. For this purpose, four motor tasks requiring finger and hand movements were implemented in a convenience sample of healthy participants to examine the system's operational modes, which allow users to play musical melodies on a touchscreen keyboard (Tasks 1–3) or draw figures/shapes (Task 4), and to assess the reliability of the device's main features such as the presentation of sequences of audiovisual stimuli at a pre-defined order and record response and movement data (e.g., reaction time, correct/incorrect responses, inter-response interval). Specifically, Tasks 1 to 3 are adaptations of motor sequence learning tasks that have been previously used in research and/or clinical practice [34, 38, 64, 65] and involve the presentation of melodies that vary in length, tempo, and complexity that are reproduced by the participant by pressing different keys represented by squares displayed on the device's touchscreen. Such tasks are often implemented in active music playing therapy to train finger dexterity, range of mobility, functional hand movements, spatial-temporal control, and limb coordination [34, 37, 39, 45, 51]. In addition, training of finger movements involving tracking a target or tracing a line on a computer screen is commonly implemented in motor rehabilitation to improve spatial-temporal control and fine motor skills of the paretic hand [66–68]. However, finger tracking training is not usually implemented in music-based interventions due to limitations imposed by the structure of the majority of acoustic musical instruments. Therefore, Task 4 is

an example of an exercise for spatial accuracy training of continuous motions via sonification, whereby the position and movement of the finger are captured in real-time and transformed into different sounds.

## Materials and methods

The experimental procedures conformed with the Declaration of Helsinki and were approved by the Local Ethics Committee of the Faculty of Psychology and Movement Science of the Universität Hamburg. All participants were fully informed about the nature of the study and provided written informed consent to participate. The individual in this manuscript has given written informed consent (as outlined in PLOS consent form) to publish these case details (Fig 4).

### Device hardware and software

The Sonification Arm Training Apparatus (SONATA) consists of a custom-made graphical user interface generated by a single-board plugged-in computer (Raspberry Pi 2 B with HiFi-Berry) displayed on a 32" touchscreen (iiyama ProLite T3234MSC-B3X; visible screen size: 698.4 x 392.8 mm; resolution: 1920 x 1080 pixels, pixel spacing: 0.364 x 0.364 mm) with built-in speakers and controlled wirelessly via Wi-Fi by a battery-powered computer tablet (Acer One 10) (Fig 1). The hardware and software of the system have been designed to minimize any latency ($\leq$ 30 ms) between user input and sound output.

### Device design, input and settings

The touchscreen user interface is programmed through a controller tablet to individualize the therapist's and the patient's work surfaces. This allows the therapist to use the controller tablet pc to program the graphic and acoustic settings for a new exercise (e.g. screen layout, sound sequences, metronome setting, drawing exercises) while the client performs a different exercise on the SONATA touchscreen interface. The number and the order of the training exercises can also be designed and saved by the therapist in the device's memory before a training session.

The graphical interface displayed on the controller tablet allows the therapist to choose between two operational modes (keyboard and drawing) and displays two distinct functions: Input and Settings (Figs 2 and 3).

The Keys Mode allows the user/patient to play sound sequences on a touchscreen keyboard by pressing different keys represented by squares displayed on the device's touchscreen. Each keypress produces a feedback sound that corresponds to a pitch. The default input window

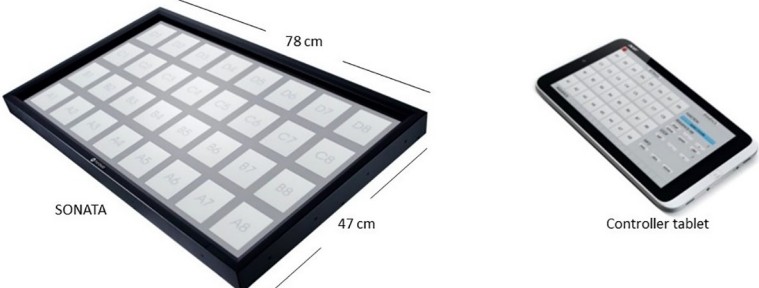

**Fig 1. Device's touchscreen, graphic user interface, and controller tablet.**

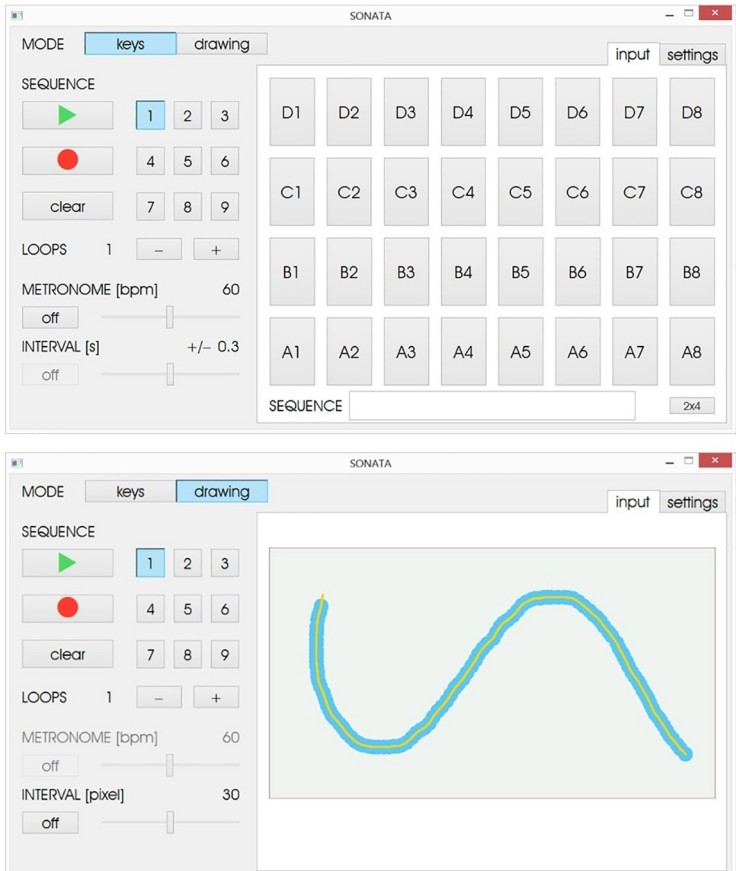

**Fig 2.** User interface input function for the keys mode (upper panel) and drawing mode (lower panel).

presents a 4x8 key matrix (4 rows, 8 keys per row) and each key can be tuned in ascending semitones (from left to right) or in diatonic scales either as a single pitch or as a triadic major or minor chord. To accommodate for less precise reaching motion, the key sizes can be increased and displayed in a 2x4 key matrix. In the input function, the therapist can program and save up to 9 sound sequences by pressing the 'record' button and then playing/pressing the sequence of keys in the required order (Fig 2). Additionally, it is possible to repeat the sequence of tones using the loop function and set the metronome tempo in beats per minute (BPM) to which the participant will synchronize their movements. Features such as the loop function and the metronome can be disabled at the discretion of the therapist according to the exercise objectives and the patient's needs.

The settings function in the Keys Mode (Fig 3) provides additional setup options where the therapist is able to select, for instance, the volume and the instrumental timbre of the feedback sounds (e.g., piano, organ, violin, guitar, trumpet, and saxophone). The sustain function determines the duration (in seconds) that the sound remains present between key presses, and the marker function enables/disables the color-highlighting function of the visual display, where each key of the sequence changes color as it is presented. During the exercise, the predefined sequence of keys is displayed to the patient by turning blue in a cumulative order and simultaneously presenting the corresponding pitch. After the sequence is introduced, the patient then reproduces the sequence of keys in the correct order and in synchrony with the metronome

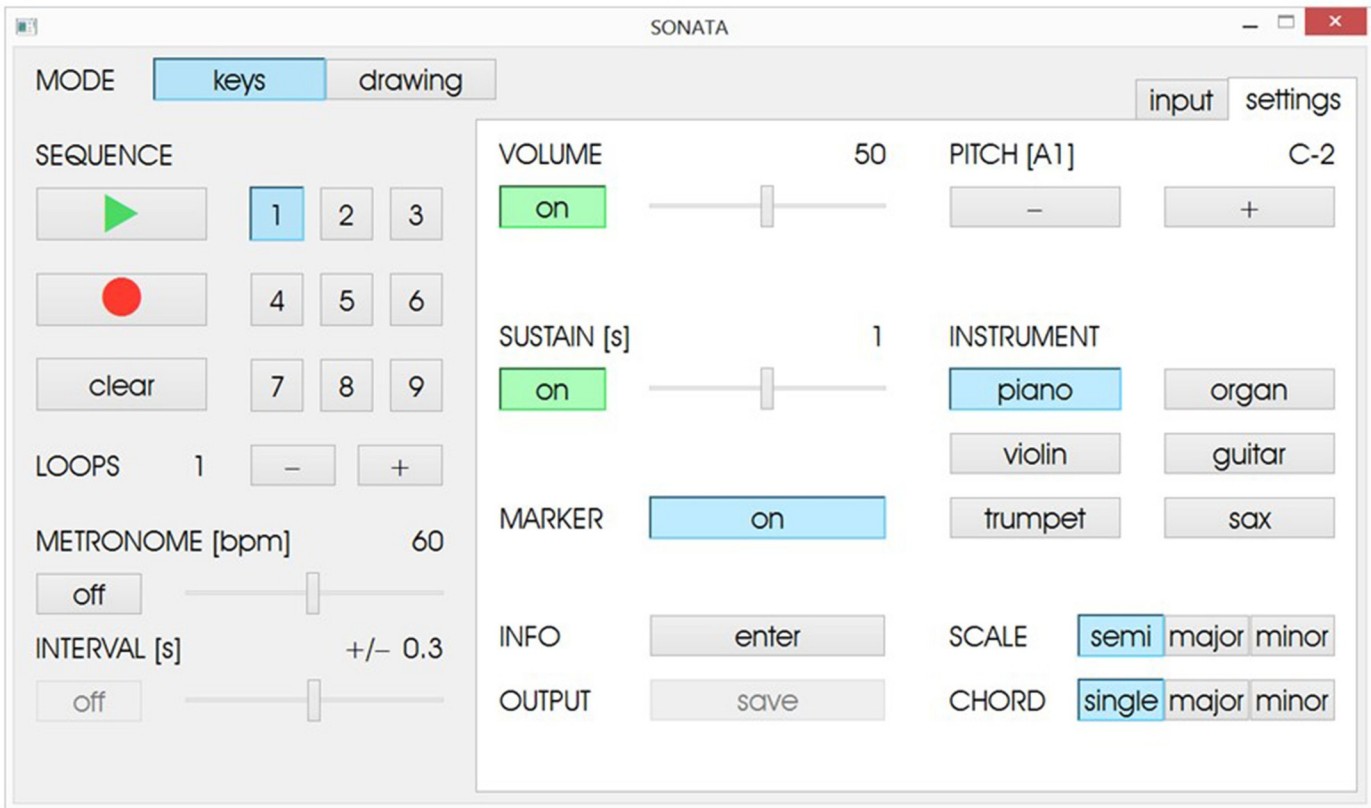

**Fig 3. Settings function for the keys mode.**

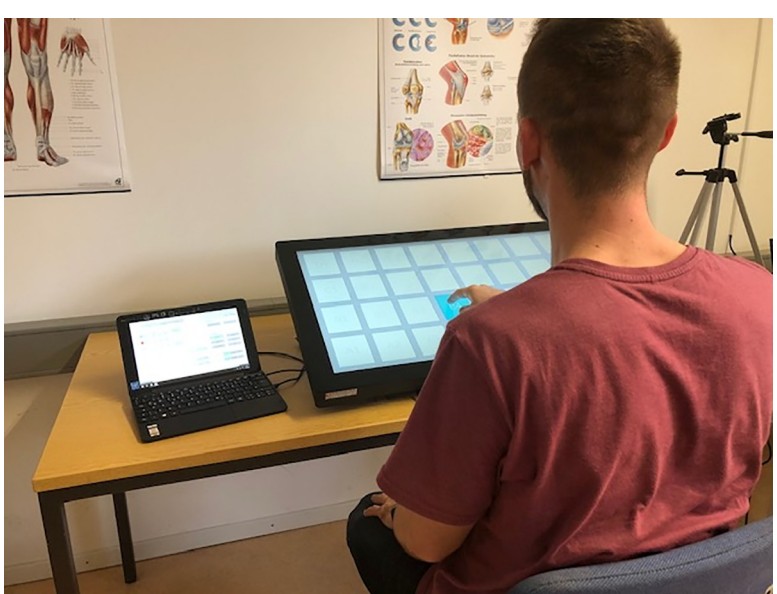

**Fig 4. Device in use during the evaluation procedures.** The individual in this manuscript has given written informed consent (as outlined in PLOS consent form) to publish this photograph.

tempo. The device also includes a function where the therapist can determine a temporal window around the metronome tempo in which the patient is required to press the keys. With this metronome interval function, if the keypresses occur outside of the predefined temporal interval, no auditory feedback is provided, thus encouraging the patient to maintain temporal accuracy.

In the Drawing Mode, the spatial accuracy of continuous motions is trained via sonification. The therapist can program up to 9 distinct figures or shapes that are subsequently traced by the patient using his/her finger. The default input window in the drawing mode presents an empty field into which the therapist can draw the figure/shape by touching the screen and moving the finger in the required direction (Fig 2). Along with the movement of the finger on the screen, a continuous sinusoidal tone is presented whereby the pitch is determined by the position of the finger on the screen, with lower tones presented on the lower quadrants and higher tones on the top quadrants of the screen in ascending order from left to right and from bottom to top. Additionally, a visual guide is displayed during the drawing where the yellow line serves as a template indicating the movement trajectory to be performed and the blue frame sets the interval in pixels in which the patient has to move the finger on the screen to train spatial accuracy. During the exercise, the figure is first displayed to the patient at the same velocity and trajectory that were recorded by the therapist, and then the patient reproduces the drawing at their preferred tempo. During the drawing, the patient's finger movement also produces a sinusoidal feedback sound that changes in pitch depending on the finger's position on the screen. It is also possible to define an area of spatial accuracy around the figure lines whereby no sound feedback is provided when the finger trajectory is outside of the predefined area.

## Data output

Tasks performed on the device automatically generate quantifiable movement data that are captured and stored for further analysis. The registered information about the user's interaction with the device during the performance of a given task can be used by therapists to assess progression through therapy as well as in empirical research.

The recorded data are stored as text files (American Standard Code for Information Interchange, ASCII) and include general information about the session (date, time, therapist), patient (ID code), and task settings such as the operational mode (keys/drawing), metronome tempo (BPM), inter-stimulus interval (i.e. the time interval between the offset of one stimulus and the onset of next stimulus (ms)), key numbers of the recorded sequence (1 to 32, from bottom left to top right), and inter-response interval (i.e. the time interval between successive keypresses (ms)). Session and patient information are included in the system at the discretion of the therapist while task-related information is derived from the exercise settings as programmed by the therapist. The Keys Mode includes analyzable data such as expected sequence key and patient response key (from 1 to 32), inter-stimulus interval, inter-response interval, and synchronization error (i.e. the phasic relationship between stimulus event and motor response). The output data provided in the Drawing Mode includes position information such as the x- and y-coordinates of the patient input (in pixels, 0 to 1880 and 0 to 1040 respectively), x- and y-coordinates of the nearest target section, and the distance between patient input and the nearest target section. From the data of the Drawing Mode, the speed of the hand/finger movement can also be derived providing a velocity profile of the drawing movement. The recorded information is stored in the controller tablet password-secured internal memory and cannot be transmitted as the device is not connected to a network.

## Evaluation procedure

**Participants.** Functional tests were conducted with 21 healthy individuals (13 males, 8 females) recruited at the Faculty of Psychology and Movement Science at the Universität Hamburg/Germany. Participants were on average 26.4 years old (*SD* = 3.5, range 21–36 years), 3 of them indicated a preference for the left hand and 18 for the right hand. All participants reported normal hearing, normal or corrected to normal visual acuity, and had no ongoing musculoskeletal injuries and no neurological damage or disorders that could influence normal upper limb movement.

**General procedures.** Participants were seated in a regular chair in a quiet test room with the SONATA device placed on a table positioned at a comfortable distance in front of them at the wrist level (Fig 4). Stimuli presentation and data collection were implemented with the SONATA device running a built-in custom-made software. Before each task, participants received written instructions and were allowed to practice the tasks. With breaks, the session took approximately 50 minutes to be completed.

**Tasks.** *Task 1*: *Serial reaction time*. The Serial Reaction Time (SRT) task evaluates motor sequence learning whereby participants are required to respond as rapidly as possible to targets (auditory and/or visual stimulus) that are presented either in a repeating order (sequence blocks) or in random order (random blocks) [64, 69–71]. Findings of studies using the SRT task consistently show a decrease in reaction time in the sequence blocks in relation to the blocks where targets are presented in random order, indicating an effect of implicit motor sequence learning [64, 69–71]. Given the robustness of the results reported in SRT studies, this task was implemented to examine some of the device's main features, such as the presentation of sequences of stimulus at a pre-defined order and the data acquisition of response information including response accuracy and reaction time.

In the present task, the stimuli consisted of visual targets (squares representing different pitches) displayed on a horizontal array of 32 locations presented on the screen (Fig 1). The visual targets were a light grey color. During each experimental trial, one target at a time was presented by turning blue while the corresponding auditory stimulus (piano tone of 1000 ms duration) was simultaneously presented. The target remained on the screen until the correct response was made. The participant's task was to reach the preferred arm and press the button corresponding to the target as fast and as accurately as possible. Stimuli were presented at a fixed 1-second interval and were divided into separate trials. In the random (R) trials, sequences of stimuli followed an unpredicted order. The random sequences were generated using a random number generator whereby each of the keys corresponded to a number from 1 to 32, starting from the top left (D1) to bottom right (A8). In the sequence trial (S), the same sequence of 12 tones was repeated throughout the task. In total, the task consisted of 8 trials: 4 random trials and 4 repetitions of the sequence trial, following a standard block design (RSRSSRRS).

*Task 2*: *Unimanual sensorimotor synchronization task*. The ability to learn new action sequences is known to be affected by task parameters such as sequence length, rate, and complexity [72, 73], as well as by individual differences in working memory capacity [74]. It has been well-documented that this learning ability is significantly impaired in aging [75–77], stroke [78, 79], and neurologic disorders [80].

In standard Music-support Therapy protocols [34], for instance, electronic keyboard and/or drum pads are used to exercise fine and gross movements whereby patients may start playing simple sequences that vary in the number of tones, movement velocity, and limb patients are required to play, which progressively increase in difficulty. Clinical studies indicate that motor improvements can be achieved already during the first training sessions with observable

changes in movement velocity, key pressure, and note accuracy [30]. Music-based exercises, such as those requiring finger dexterity using electronic keyboards [34], might be adapted to touchscreen devices [29].

Therefore, the following tasks (unimanual and bimanual sensorimotor synchronization) were implemented to test the utility of the device's Key Mode and the reliability of the data automatically acquired by the system in tasks that are often implemented in fine motor training. This task evaluated the Keys Mode of the device, which allows the presentation of pre-defined sequences of melodies that are reproduced by the user by reaching the preferred arm to press different keys represented by squares displayed on the device's screen. For that, participants were presented with 8 distinct pre-defined melodies composed of six to nine tones. Light grey squares representing 32 distinct pitches were displayed on the SONATA screen. During each trial, one note of the melody at the time was presented on the screen by turning blue while the corresponding auditory stimulus (piano tone) was simultaneously presented and remained on the screen cumulatively. Melodies were designed to follow different motion patterns on the screen and differed in relation to sequence length (6–9 tones) and inter-stimulus interval (slow: 66 BPM/910 ms; fast: 80 BPM/750 ms). Participants were instructed to memorize the order of each note of the melody and then reproduce the sequence in the correct order and in synchrony with the metronome using their preferred hand. Each melody was presented only once (total of 8 trials).

*Task 3*: *Bimanual sensorimotor synchronization task*. Similar to Task 2, a set of 9 pre-defined melodies composed of seven notes was presented. During each trial, one note of the melody at a time was presented on the screen by turning blue while the corresponding auditory stimulus was simultaneously displayed. Each note of the melody was presented at a fixed tempo (BPM 80/750 ms). The participants' task was to memorize the order of each note of the melody and reproduce the sequence in the correct order and in synchrony with the metronome. Additionally, participants were instructed to reach the arm and press the target notes appearing on the left side of the screen with the left hand, whereas notes appearing on the right side had to be played with the right hand. Each melody was only presented once, totaling 9 trials.

*Task 4*: *Finger tracking task*. Hand and finger functions are often impaired in neurologic disorders such as cerebral palsy [81], stroke [82], and Parkinson's Disease [83], with a significant impact on tasks that require fine motor control, including drawing and finger tracking [66, 84]. Movement training to enhance hypometria (i.e. lack of motor coordination where movements fail to reach the intended target), slowness of movement (bradykinesia), and weakness are important treatment targets for neurologic rehabilitation [66, 82, 84]. Training of finger movements involving tracking a target on a computer screen with reciprocal extension and flexion of movement of the index finger has been previously applied in motor rehabilitation [66–68], with results suggesting significant improvements in tracking accuracy with transfer of gains to grasp and release function [66].

This functional task was implemented to assess features of the Drawing Mode of the device including movement sonification, the presentation of different target waveforms or shapes, as well as the acquisition of movement data relating to accuracy and time for completion. In this task, participants were instructed to follow two different figures displayed on the screen. One of the figures consisted of a sine wave shape (92 cm length) while the second figure was a triangle wave shape (109 cm length) that was displayed horizontally throughout the entire screen. Participants' task was to follow the shape displayed on the screen with the index finger of their preferred hand moving from left to right at their preferred rate using flexion-extension movements at the elbow and shoulder, repeating the task 5 times per trial. In some trials, the movement of the finger on the screen generated auditory feedback (sinusoidal tones) that changed in frequency as the finger moved upward (higher frequency) or downward (lower frequency),

whereas no auditory feedback was presented in half of the trials. Figures were presented in separate blocks with counterbalanced order. Each block consisted of 8 trials; 4 trials with auditory feedback and 4 trials without auditory feedback, totaling 16 trials.

## Statistical analysis

In Task 1, the main variable of interest was the mean reaction times for each condition (random and sequence trials). Absolute reaction time was evaluated across conditions with a univariate analysis of variance.

In task 2, we were interested in whether the number of errors (i.e. pressing the wrong note) would differ in relation to the sequence length (6 to 9 tones) and rate (slow and fast). To obtain the information regarding accuracy, we compared the sequence note presented with the participant's actual response, converted in percentage. A two-way analysis of variance was performed with the percentage of correct responses as the dependent variable and sequence length (4) and rate (2) as factors. Additionally, we assessed whether participants were able to synchronize their movements with the metronome, using the mean and standard deviation of the inter-response interval (IRI).

In Task 3, participants performed the task bimanually, receiving instructions to press the target notes displayed on the left side of the screen with the left hand and the notes displayed on the right side with the right hand. We were interested in whether performing the task bimanually would affect accuracy (i.e. number of errors). The percentage of correct responses was obtained by comparing the information regarding the notes presented with the participant's actual keypresses, while data on lateralization errors were recorded manually by the experimenter. Descriptive statistics are presented. Additionally, we also assessed whether participants were able to perform the task following the tempo set by the metronome (mean/standard deviation of IRI).

In the finger tracking task (Task 4), the variables of interest were the time needed to complete each trial and drawing accuracy. The information regarding time for completion was obtained by the sum of the time difference between two points on the screen in milliseconds (screen resolution 1920 x 1080 pixels). Drawing accuracy was computed comparing the distance between the template figure and the participant's drawing. We were interested in whether the time for completion and drawing accuracy would be affected by the shape of the figure and the availability of auditory feedback generated by the movement of the finger on the screen. A multivariate analysis of variance was performed with time (seconds) and drawing accuracy (pixels) as the dependent variable and with auditory feedback (with and without) and figure shape (sine wave or triangle wave) as factors.

For all statistical comparisons, the significance level was set to 5% ($p < 0.05$). Statistical analysis was performed using SPSS 24.0 (SPSS Inc., Chicago, IL, USA). The de-identified data (S1 Dataset) and metadata (S1 File) are available as supplementary material.

## Results

### Task 1: Serial reaction time task

Participants performed the task with an average of 100% accuracy ($SD$ = 1.7%), demonstrating that they were able to reach the correct target position in both conditions (random and sequential). The analysis of the mean absolute reaction time indicated that participants were significantly faster to respond in the sequential order ($M$ = 494 ms, $SD$ = 40 ms) than in the random order trials ($M$ = 510 ms, $SD$ = 55 ms; $p$ = 0.03). These results concur with previous studies showing a decrease in reaction time during the sequence trials in relation to the random trials, which is indicative of implicit motor sequence learning [64, 69–71].

## Task 2: Unimanual sensorimotor synchronization task

In this task, we were interested in whether accuracy would be affected by sequence length and presentation rate. Overall, participants performed the task with an average of 96% accuracy ($SD$ = 10%). Nonetheless, the analysis indicated that there were significant main effects of sequence length ($F(3,140)$ = 5.896, $p$ = 0.001) and rate ($F(1,140)$ = 11.036, $p$ = 0.001) on the percentage of correct responses, but there were no significant interaction between factors ($p$ = .607). Further comparisons with Bonferroni corrections indicated that sequences with 9 tones had significantly more errors than sequences with fewer tones ($p < 0.05$), and that sequences presented and performed at a faster rate had more errors than sequences at a slower tempo ($p$ = 0.02). These results corroborate findings consistently reported in previous studies demonstrating that accuracy can be affected by task parameters such as sequence length, rate, and complexity [72, 73], which is indicative that the data recorded by the device is reliable. Analysis of the IRI showed that participants were able to synchronize their movements according to the metronome tempo, as the average IRI during the slow sequences was 897 ms ($SD$ = 153 ms) and during the fast sequences the average IRI was 740 ms ($SD$ = 77 ms).

## Task 3: Bimanual sensorimotor synchronization task

Overall, the task was performed with an average accuracy of 85% ($SD$ = 20%), suggesting that performing the melodic sequences with both hands resulted in an increased number of errors (i.e. pressing the wrong note). When considering lateralization errors, the average accuracy was 99% ($SD$ = 2.8%), demonstrating that participants were able to perform the task using the correctly assigned hand. Finally, the analysis indicated that participants performed the task with an average of 791 ms inter-stimulus interval ($SD$ = 223 ms), thus significantly slower ($t$ (166) = 2.394, $p$ = 0.018) than the tempo set by the metronome (BPM 80/750 ms).

## Task 4: Finger tracking task

In task 4, participants had to track with their index finger distinct shapes following a template displayed on the screen. We were interested in whether the time for completion and drawing accuracy would be affected by the shape of the figure and the availability of auditory feedback generated by the finger movement on the screen. Statistical analysis indicated that there were no significant interactions or main effects of figure shape or auditory feedback condition on time and drawing accuracy. Drawing accuracy, as measured with the mean distance between the participants' finger trace and the figure template (in pixels), did not differ significantly in the sine wave shape ($M$ = 22.5, $SE$ = 6.2) and triangle wave ($M$ = 22.3, $SE$ = 6.2, $p$ = 0.98). When considering the effect of the availability of auditory feedback, mean drawing accuracy was not significantly different in the auditory feedback condition ($M$ = 29.25 pixels, $SE$ = 6.2) and with no feedback ($M$ = 15.58 pixels, $SE$ = 6.2, $p$ = 0.12). Time for completion also did not differ significantly between sine wave ($M$ = 46.6 sec, $SE$ = 28.4 sec) and triangle wave ($M$ = 68.9 sec, $SE$ = 28.4 sec, $p$ = 0.58), and between trials with auditory feedback ($M$ = 84.6 sec, $SE$ = 28.4 sec) and without auditory feedback ($M$ = 30.9 sec, $SE$ = 28.4 ms, $p$ = 0.18).

## Discussion

In this study, we describe a novel music-based therapeutic device for upper extremity movement training called SONATA and evaluate the system's functioning and usability in a convenience sample of healthy individuals. Four motor tasks requiring finger and hand movements previously used in research and/or clinical practice [34, 38, 64–66] were adapted to test the device's operational modes (keyboard and drawing) and main features.

Overall, the present pre-clinical trial indicates that the device's hardware and software reliably present pre-defined sequences of audiovisual stimuli and capture and store response and movement data. For instance, the results of the functional tests concur with the findings consistently reported in previous research, indicating that the data recorded by the device is reliable. In Task 1, results indicated a decrease in reaction time in trials where targets are presented in a repeating order compared to random order, a finding that has been consistently found by studies using the Serial Reaction Time paradigm [64, 69–71]. The results of Tasks 2 and 3 concur with the notion that sequence length, rate, and complexity significantly affect the accuracy of newly learned action sequences [72, 73], as our findings indicated that sequences with more elements or presented at a faster tempo had significantly more errors than sequences with fewer tones or slower rates, and that performing melodic sequences bimanually resulted in an increased number of errors. Task 4 revealed that healthy participants' were equally accurate in tracking different waveforms independent of movement sonification conditions [66–68].

Our results also indicate that the device is feasible and easy to use by healthy individuals. This was demonstrated given the observation that participants were able to access and complete all tasks using the tested device with minimum assistance. Participants received written instructions on how to perform the exercises and were allowed to practice each task to ensure that they understood the instructions and were able to follow the procedures. Participants indicated that they felt confident to perform the tasks after a single practice block and did not require further support during the experimental trials (e.g. verbal reminder of the instructions or demonstration on how to perform the tasks). The participant's performance accuracy further suggests that task difficulty levels were appropriate for healthy young adults. However, we acknowledge that these exercises may be cognitively demanding for neurologic patients. Thus, task features such as sequence length, rate, and complexity might need to be adjusted when implementing similar exercises on individuals with important cognitive impairment. This observation also applies to the tasks requiring movement synchronization to a metronome, which may be difficult to perform depending on the severity of the motor and/or cognitive impairment, type of injury, or stage of the condition (acute, sub-acute, chronic). The therapist may opt to adjust the metronome tempo, use the 'metronome interval' function to increase the temporal window around the metronome tempo in which the patient is required to press the keys, or completely disable the metronome depending on each individual's needs. When implementing training exercises with individuals with neurologic disorders, therapists may use rhythmic auditory cues, such as a metronome, to facilitate movement planning and execution through auditory-motor entrainment to reduce reliance on stereotypical compensatory upper extremity movements (e.g. trunk flexion, excessive shoulder abduction, circular arm movements) and provide additional manual assistance to fixate trunk position during the exercises [44, 46].

In the past years, researchers and clinicians have acknowledged the relevance of technology to open the possibility of people without music training to engage in active music playing, to facilitate access to music-based therapy in different settings, and to increase motivation and client participation, while offering professionals the opportunity to deploy more resources to meet the patient's treatment goals [60]. However, there is a need for the development of hardware and software specifically designed for clinical practice. This pre-clinical study focused on describing and testing the functionality and usability of a device designed for upper extremity motor function rehabilitation with healthy individuals. Hence, further research is needed to examine the feasibility, ease of use, and reliability of the data acquired by the system in clinical studies with neurologic patients.

## Future research

The device described and evaluated in the present study may be a potential tool for the implementation of music-based interventions in neurologic rehabilitation. All tasks administered in this study required the presentation of pre-defined sequences of audiovisual stimuli or shapes and were designed to include different sequence lengths, presentation rates, task complexities, changes in the availability of real-time auditory feedback as well as capturing distinct response and movement data. These are all aspects carefully considered in the implementation of individual therapeutic plans, thus indicating that the device may be a useful tool to assist therapists during treatment implementation and assessment of targeted therapeutic goals. To examine the feasibility of this device in a clinical context, further research is needed to evaluate therapists' interactions with the device including usability questionnaires and a qualitative assessment of the device's interface design, ease of use, safety, feasibility, and versatility.

In this study, we adapted tasks often used in music-based interventions to train fine motor skills using electronic keyboards [30, 34] and touchscreen devices [40], as well as finger tracking tasks implemented in motor rehabilitation that involve tracing a target on a computer screen [66]. These tasks are simplified examples (and not an exhaustive list) of the different types of training exercises that could be used in neurologic rehabilitation with the device tested here. However, adjustments may be needed to use these tasks with neurologic patients in future clinical research. For instance, our findings indicated that sequence length, presentation rate, and the number of limbs required (unimanual or bimanual) can significantly influence accuracy in motor sequence learning tasks. These results, thus, suggest that these aspects need to be thoughtfully considered by researchers and therapists when planning similar exercises with cognitively impaired patients. Participant's cognitive function, including attention, short-term memory, and executive function, should be assessed to better guide the therapist/ researcher in selecting and designing the tasks according to the individuals' capacity. This also applies to bimanual versus unimanual training [85]. Moreover, therapeutic protocols often implement training exercises that vary in length, movement velocity and direction, and these aspects progressively increase in difficulty [30]. Here, for research purposes, we combined all these features to demonstrate the different device settings that can be programmed and incorporated in the design of training exercises. However, it must be noted that all these aspects are precisely selected in a treatment protocol upon thorough assessment so that the training is adequate to the patient's treatment goals and objectives, providing exercises that challenge the patient but that are achievable, promoting autonomy and giving patients control over the learning experience.

Future clinical research may also benefit from the evaluation of whether the data recorded with this device in tasks involving continuous motions is accurately captured when the tasks are performed by patients with significant movement impairment, such as apraxia (i.e. a motor disorder that affects a person's ability to execute movements when asked to), tremor and bradykinesia. The level of assistance required by patients to perform different rehabilitation exercises with this device should also be subject to examination to better define its feasibility depending on the severity of motor and/or cognitive impairment, type of injury, or stage of the condition. The level of support required to access and complete different training exercises with the SONATA will also help define guidelines for the use of this device in different clinical settings (hospitals, care homes, or self-implemented home training).

Finally, research is warranted to better investigate the potential use of interventions based on motor learning to impact cognitive and emotional domains aside from the expected motor improvements. Active music playing as a therapy is an enjoyable activity that involves complex and coordinated movements while placing a high demand on cognitive functions, such as

attention, working memory, and executive function [27, 86]. It has been demonstrated that active music engagement through instrument playing promotes significant cognitive benefits to attention and verbal memory [87, 88]. Moreover, several neurologic music therapy interventions have been developed specifically for cognitive rehabilitation focusing on auditory attention and perception training, memory training, and executive function training [89–91]. Future clinical research would be of interest to test whether interventions for cognitive rehabilitation could also be implemented with the SONATA.

Clinical studies are currently in place to test the feasibility and usability of the device in upper extremity movement training. Specifically, the Keys Mode of the SONATA was used to implement a 3-week Therapeutic Instrumental Music Playing (TIMP) protocol to train patterns of reaching movements involving wrist flexion, elbow flexion/extension, shoulder flexion/abduction/adduction, and trunk rotation in chronic stroke patients. The study results supported the effectiveness and feasibility of the SONATA as a music-based device to enhance motor recovery in stroke rehabilitation [92].

## Limitations

This pre-clinical trial was restricted to testing the hardware/software of a novel music-based therapeutic device with highly educated, healthy, young adults. Thus, the results of this study may not directly apply to a clinical patient population. Further research on user experience evaluation from both patients and professionals are needed to examine the feasibility, usability, and ease of use of the device in different clinical settings. The feasibility of the exercises used in this study also needs to be tested with clinical patient populations requiring neurorehabilitation, with data captured with the device being correlated with standard measures to assess the reliability, acceptability, tolerance, and adherence of treatment protocols implemented with the SONATA. Moreover, prior to the implementation of large-scale clinical trials, the device needs to undergo appropriate regulation and registration.

## Conclusions

A novel music-based therapeutic device called SONATA was presented and tested in the present pre-clinical, single-arm trial. Four motor exercises requiring finger and hand movements previously used in research and/or clinical practice were adapted to test the device's functioning and usability with healthy young adults. The results of the functional tests suggest that the device is a reliable tool to present pre-defined sequences of audiovisual stimuli and shapes and to record and store response and movement data, such as reaction time, correct/incorrect responses, inter-response interval, and movement spatial accuracy. In addition, this preliminary study suggests that the device is feasible and adequate for use with healthy individuals.

The findings presented here open new avenues for further clinical research to investigate the usability of the SONATA for the implementation of upper extremity motor function training in neurological rehabilitation. We also discussed directions for future research, which include further user's experience evaluation from both the therapists' and the patients' perspectives to better understand, for instance, the level of training required to operate the device, therapists' interaction with the system to program different training exercises as well as patient's acceptability, tolerance, and adherence. Phase I clinical studies are also needed to examine the effects of training protocols implemented with the SONATA for upper extremity movement training with clinical populations in different clinical settings (hospital, care homes, community, homes) and direct the development of specific guidelines and training protocols.

## Supporting information

**S1 Dataset. De-identified datasets.**
(PDF)

**S1 File. SONATA output variables.**
(PDF)

**S2 File.**
(PDF)

## Acknowledgments

The authors gratefully acknowledge all involved in this study, with special thanks to Sophie Platzer and Paul Weidenmüller for their assistance in data acquisition and all participants for their collaboration.

## Author Contributions

**Conceptualization:** Nina Schaffert, Michael H. Thaut.

**Data curation:** Nina Schaffert.

**Formal analysis:** Thenille Braun Janzen.

**Investigation:** Nina Schaffert.

**Methodology:** Nina Schaffert, Thenille Braun Janzen, Sebastian Schlüter, Michael H. Thaut.

**Project administration:** Nina Schaffert.

**Resources:** Nina Schaffert.

**Software:** Roy Ploigt, Sebastian Schlüter.

**Supervision:** Michael H. Thaut.

**Validation:** Thenille Braun Janzen, Michael H. Thaut.

**Writing – original draft:** Nina Schaffert, Thenille Braun Janzen.

**Writing – review & editing:** Nina Schaffert, Thenille Braun Janzen, Sebastian Schlüter, Veronica Vuong, Michael H. Thaut.

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
