## [Decision Letter · Decision Letter 0]

9 Sep 2020

PONE-D-20-23749

Development and Evaluation of a Novel Music-Based Therapeutic Device for Upper Extremity Movement Training

PLOS ONE

Dear Dr. Schaffert,

Thank you for submitting your manuscript to PLOS ONE. After careful consideration, we feel that it has merit but does not fully meet PLOS ONE’s publication criteria as it currently stands. Therefore, we invite you to submit a revised version of the manuscript that addresses the points raised during the review process.

ACADEMIC EDITOR: As also one of the reviewers stated, I believe your study might be a valuable step towards adding new tools to help people with motor impairment improve their function.  In this context, I thank you in advance for your careful revision work in line with the points, comments and suggestions highlighted by our reviewers in order to make your article more clearer and more useful for interested researchers.

We look forward to receiving your revised manuscript.

Kind regards,

Sukru Torun

Academic Editor

PLOS ONE

Journal Requirements:

2. Thank you for including your competing interests statement; "The authors have declared that no competing interests exist."

We note that one or more of the authors are employed by a commercial company: BeSB GmbH

3.  We note that Figure [4] includes an image of a [patient / participant / in the study]. 

4. 

We note that you have indicated that data from this study are available upon request. PLOS only allows data to be available upon request if there are legal or ethical restrictions on sharing data publicly. For information on unacceptable data access restrictions, please see http://journals.plos.org/plosone/s/data-availability#loc-unacceptable-data-access-restrictions.

Additional Editor Comments (if provided):

Dear authors,

As also one of the reviewers stated, I believe your study might be a valuable step towards adding new tools to help people with motor impairment improve their function. In this context, I thank you in advance for your careful revision work in line with the points, comments and suggestions highlighted by our reviewers in order to make your article more clearer and more useful for everyone interested.

Reviewers' comments:

Reviewer's Responses to Questions

**Comments to the Author**

1. Is the manuscript technically sound, and do the data support the conclusions?

Reviewer #1: Partly

Reviewer #2: Partly

2. Has the statistical analysis been performed appropriately and rigorously? 

Reviewer #1: Yes

Reviewer #2: I Don't Know

3. Have the authors made all data underlying the findings in their manuscript fully available?

Reviewer #1: No

Reviewer #2: No

4. Is the manuscript presented in an intelligible fashion and written in standard English?

Reviewer #1: Yes

Reviewer #2: No

5. Review Comments to the Author

Reviewer #1: Data Availability: The form states, “Stating ‘data available on request from the author’ is not sufficient. If your data are only available upon request, select ‘No’ for the first question and explain your exceptional

situation in the text box.” However, the authors stated, “The datasets generated and/or used during the current study are not publicly available but are available from the corresponding author on reasonable request.” No description of an exceptional situation was provided.

Major Comments:

- Throughout much of the manuscript, I found myself asking, “How is the SONATA different from existing music therapy devices? What is the key innovation?” The manuscript does a good job of motivating music-based rehabilitation but does not discuss the gaps that the SONATA addresses until the end of the manuscript. An earlier, structured review of relevant devices would be useful to motivate the development of the SONATA and clarify the message of the manuscript.

- Is there a reason the grid layout was chosen as opposed to a typical keyboard layout? Were other layouts tested, and, if so, did they have any impacts on the results? I also have similar questions regarding the size and shape of the virtual keys. (e.g., Can smaller keys be used to encourage greater spatial accuracy?)

- Introduction, last paragraph: “With that in mind, a novel music-based therapeutic device for upper extremity movement training was developed to improve upper extremity motor function, to increase independent patient engagement, to enhance treatment quality, intensity, and compliance, and to assist therapists during treatment implementation and assessment.” This sentence suggests that the SONATA does these things, but they have not been validated and this study does not attempt to demonstrate them. The literature review suggests that these claims could be true for the SONATA, but the statement in its current form is misleading.

- Introduction, last paragraph: “to evaluate the feasibility of the system’s functioning for upper extremity movement training.” Given the subject set, this statement is also misleading.

- Methods, Section 2.3: “… thus assisting the therapist in assessing measurable changes in upper limb function throughout training.” This claim may need more justification. The experiment does not show that the data collected by the tasks, as they are implemented by the SONATA, correspond to changes in upper-limb function. How the therapists interact with this data (e.g., visualization, as opposed to raw data) is also important to determining if the data assists therapists. If it is too difficult to draw actionable inferences from the data, then in practice it will just be ignored.

- Discussion, second paragraph: Is performance accuracy really an appropriate measure of ease of use? How hard does a device have to be to use to elicit a significant decrease in accuracy for young, healthy subjects? It also does not account for the interface design or therapist interactions with the system. Some sort of usability questionnaire or qualitative assessment may help to better support this claim. The limitations paragraph mentions the need for additional evaluation, but this statement is too strong.

- Conclusions: “… indicate the feasibility and reliability of the device as a tool for upper extremity movement training.” This statement is too strong given the subject demographics and experimental design. More cautious language should be used.

Minor Comments:

- Methods, Section 2.1: Please include a more thorough description of the hardware. For example, is the device battery-powered or plugged in? Is the system wirelessly controlled via Wi-Fi, Bluetooth, or some other standard? If the device was designed to minimize latency between user input and sound output, what was the average latency?

- Figure 1: It would be useful to readers to visually convey the scale of the devices. From the text, I see that the SONATA is rather large (approx. 0.7 m by 0.4 m), but it appears much smaller in this figure because of the adjacent tablet. (Though Figure 4 helps with this issue to some extent.) The figure should also more clearly identify which device is the SONATA and which is the tablet to prevent potential confusion.

- Figure 2: The authors may want to consider splitting or vertically arranging this figure so that the interfaces are easier to see. In comparison, Figure 3 was much easier to see.

- Methods, line 160: “BPM” Please define abbreviations before use for clarity. Readers that are not familiar with music may not know that this means beats per minute.

- Methods, Section 2.4.1, line 224: This is admittedly a bit of a nit-pick (i.e., the statement is acceptable as-is), but neurological damage or disorders can also affect upper limb movement.

- Methods, Section 2.4.4: For clarity and readability, please avoid using variable names (e.g., “PAT_TIME”) without first defining them in English. At the minimum, a table of variable names and their descriptions could be provided and referenced before use. Avoiding the use of variable names altogether and using English instead is even more preferable.

- Results, Task 4, lines 407–410: Is ms an appropriate unit here? Reporting seconds rounded to one significant digit after the decimal would be more easily parsed by the reader and would not change the conclusions.

Spelling and Grammar: Below are a few miscellaneous items the authors may wish to address. This list is not necessarily comprehensive; the authors should do another proofreading pass.

- Introduction: “Post-stroke” is usually used as an adjective and not as a noun. The authors may want to consider saying “stroke” or “stroke survivors”.

- Introduction, line 112: “With that in mind,” The “that” is ambiguous.

- Methods, line 130: “built-in”

- Methods, line 250: “pitches”

- Methods, line 282: “composed of six to nine”

- Methods, line 294: “composed of seven”

- Methods, line 295: “at a time”

- Decimal values should have a leading zero where appropriate. (e.g., “p < 0.05” on line 360.)

Reviewer #2: You have described a tablet device designed for upper limb training and presented data collected from participants' hand and finger movements using the device who do not have upper limb impairment (normal controls).

Please add to the title 'a pre-clinical, single arm trial' I would also consider revising the title further, to specify hand and finger or fine motor training, since there is no description in the manuscript of elbow or shoulder extension/flexion, aduction/abduction, or other gross motor movements.

In the abstract you use the terms neurologic recovery and neurologic conditions, please be more specific.

Line 32, correct to 'built-in'

Line 46, Abstract, this conclusion cannot be justified. please amend to reflect your findings from pre-clinical phase, i.e. it is not known whether this would work with patients who have UL paresis or apraxia.

Line 55, just state 'Stroke'

Line 74: References 35, 37, 38, 39 are not TIMP as they do not require facilitating music for movement synchrony.

Line 82. It is not clear what you are saying here. It seems that you are referring to other research here (Rojo et al, 2011, Ripolles et al, 2015, Altenmüller et al 2009, etc), where audio-motor coupling is discussed, which you cite later. Please revise the whole literature review to make it shorter and more succint, including only the key literature relevant to UL rehab (hand, finger, fine motor) and the use of rhythm-based interventions and technology such as keyboards and touchscreen instruments.

Line 117. It needs to be completely clear what you have done thorughout the manuscript, the this is pre-clinical testing, which can only show that the hardware and software is working with normal controls.

Line 130correction 'built-in, and omit the word 'sound', just write 'speakers'

139-142 graphically and acoustically program, please re-write to explain more clearly. Do you mean that the therapist (music therapist?) programs the screen layout, metronome setting and/or drawing exercise while the patient is perfomring the exercises? Please clarify.

161: do you mean synchronise their movements? Please clarify. What if they have cognitive (attention and exec function) impairment, and find the metronome distracting, performing UL movements better without it? This needs some dsicussion later

203 Please define what this (ASCII) file is. Readers really need to know that dat collection and storage is secure and can conform to relevant policies to protect patient identity. Please clarify that your system is secure in this regard.

208 'The output files also contain timing and location information of each patient’s input.' clarify, does this mean time of the session and where it took place? If so, how are these data stored to ensure data protection?

210 'inter-stimulus interval, inter-response interval' please define

219 'Functional tests were conducted with 21 healthy individuals (12 males, 8 females) recruited at...' this amounts to 20.

228 No info here about reducing compensatory movements in people who actually have UL impairment, nor at any point later in the manuscript. This must be discussed to some degree.

268-269 youve cited MST rsearch but under the name of TIMP, this needs correcting, and here you need to reference MST to help identify what it is to the reader

288-290 why such a cognitively taxing task when the focus is on UL rehab? How is this supported as a key component for UL rehab, any literature?

308 hypometria, bradykinesia, breif definition required

309-312 thi should be in the background section

328-329 (PAT_TIME –

TICK_TIME) please define

334 SEQ_KEY and PAT_KEY please define

376 'Nonetheless, the analysis indicated that there were significant main effects of

377 sequence length' This is useful data for informing on how to adapt the software/exercises for phase 1 clinical trial. There should be some discussion of this under 'future research' or 'clinical trial planning'

379-381 Anyone working in neurorehab (occupational, physiotherapis for example) would anticipate this response and be able to set up exercises of appropriate complexity for patients, based on their assessment of cognitive function, before the session. Can you explain how this is adding to or enhancing current neurorehab provision, for example in stroke? 382-384 Needs later discussion or some acknowledgement, as patients with cognitive impairment (attention and executive function) may not be able to synch with the pulse, depending on injury or stroke type and time post onset.

387 Needs discussion in the context of how to adapt the equipment.software for clinical trials/use

395 section on finger tapping task: Obviously this needs discussion in context of clinical trials, i.e. you need to test whether the data are accurate for those with apraxia for example

419-421 please amend. feasibility and ease of use of the device are not demonstrated by participants’ performance accuracy. It is the fact that they could access and complete the exercises using this equipment (each clinical setting also needs to be taken into consideration, i.e. acute, subacute, home, carehome, etc) that demonstrates this, but only if you can describe the level of support that was required by the therapist/facilitator. Was lots of assistance required so that participants could follow the exercises? How has your trial informed on the next stage, clinical trial.

422 'we replicated the results of previous research' what research? You state this previously, in the abstract. It is not clear.

440 is cognition consistently part of the focus throughout? Please ensure you maintain focus, what is primary, secondary, i.e.you need to state from the ofset that attention, executive function (memory you mention) are important to potential patient populations being able to access these exercises, that the clnician operating and delivering must consideer these functions and that there may be benefits in these domains from patients doing these exercises, as has been found in some MST research (Grau-sanchez for example).

447-450 ' Studies have consistently demonstrated that active music playing is effective to train upper extremity movement [35-44] due to crucial elements, such as the display of real-time multisensory information [54-56] and the use of metronome or beat-enhanced music to support movement training [49,62,63], which promote neuroplasticity [39,57-61]...' The way this is written suggests that these citations (39, 57-61) involved metronome or beat synchronisation, which they didn’t so I suggest re-writing this.

454-455 ...'including the availability of features that allow for better control and documentation of the training exercises implemented in each session.' No literature is cited here that indicates this is needed and this was not a clearly stated key aim at the ofset. What data supports a claim that this equipment achieves this? Have you clearly descibed how SONATA does this? Has it been reliability checked?

458 but no correlation has been made between your data and standard assessment tools for UL function such as Fugl Meyer and ARAT? SO how do you know that this helps to achieve functional goals? PLease more clearly correlate, if possible, your data with that of MST studies where there is some correlation between tapping tasks and motor assessment outcomes.

458-459 this hasn’t been feasibiity tested though. You don't know how SONATA will perform in any setting. Please amend and state that this is a limitation of your study and/or that you need to conduct a clinical trial, most likely a feasibility study.

465 ...'transform the therapeutic process.' In the interest of not making exaggerated claims, in what context? One paper is cited hear, published in a music therapy journal. You have completed a pre-clinical testing study with normal controls, not patients. Please write in a way that does not imply that SONATA can achieve this transformation. So far, you can only use published data, primarily from MST studies.

472 Can you be clear, are you saying that studies are under way (recruiting) with clinical populations?

475 need to be specific here and say what the limitations definitely were: our study was limited to testing the software and data collection. No data were correlated with standard UL measures, which needs to be done in a formal trial.

Exercises have not been tested for feasibility of delivery with stroke or other patient populations requiring neurorehabilitation in any clinical setting. Feasibility needs to be quantified in terms of acceptability of the exercises, tolerance and adherence. Following this, if data are favourable, a pilot or definitive trial would be required with a sufficiently powered sample.

Prior to that, since the device is intended for biomedical treatment, it would need to be registered as a medical device. The latter, must be acknowledged/discussed and is not mentioned at all throughout the manuscript, can you justify why it is not mentioned? If not, plese amend.

481 '9 sequences' if you are presenting this as a potential tool to aid UL rehab, then you must discuss more about the implications. Since tolerance and adherence to treatment are central in enabling patients to achieve the required high dosage of treatment (MST = 3-4 weeks at 5 X per week) to show any clinical or statistical significance, wouldn’t 9 sequences per task fall rather short of achieving this?

487 'Upper extremity movement training' Please consider revising the manuscript to consistently state hand and finger and/or fine motor function. THink about where you want to take this research, what the next stage is. This study must support the next stage, if enough has been done here to pave the way, together with all pre-existing evidence.

488 'well known' be precise, clinically relevant. Are they well known? Where are the citations?

488 'test the feasibility, ease of use.' To a degree, but the data need to be collected alongside standard UL measures from people with UL impairment and reliability checked

492 implementation and independent patient engagement ( I assume you mean self-delivery) these are two diferent things. Please clarify and define.

The conclusion could be better written. To say that results indicate feasibility and reliability is too general, too many areas are not covered in your study. What about positioning to mitigate compensatory movements, data protection (storage and transfer of data), checking of SONATA data alongside standard UL assessment tool data, infection control of equipment in clinical settings, who would deliver the exercises, a music therapist, any clinician, what can you conclude needs to be done next based on your findings, what dosage of SONATA exercises could be delivered in a clinical setting and are there currently enough exercises, how long does it take a clinician to program each exercises for each patient and what is the impact on managing a clincal case oad, what about medical device registration for SONATA?

The study might be a useful step in the right direction (adding clinical tools to help people with UL impairment to recover function), but as the manuscript stands it is not clearly framed and needs to be amended to consistenly adhere to the main areas of focus, purpose, findings, so that it is completely clear what the implcations are for future, related research, giving some indication of if and when SONATA might become a clinical tool.

6. PLOS authors have the option to publish the peer review history of their article (what does this mean?). If published, this will include your full peer review and any attached files.

Reviewer #1: No

Reviewer #2: **Yes: **Alexander Street

---

## [Author Response · Author response to Decision Letter 0]

27 Oct 2020

Dear Dr. Torun,

We thank you and the reviewers for the constructive feedback provided and have amended the manuscript accordingly.

The main points of revision were made in the Introduction and Discussion sections to clarify aspects relating to the motivation for the development of the device and to better discuss points raised by the reviewers regarding the study results, future research and study limitations.

 We have also addressed the additional journal requirements in relation to manuscript style and amended the Funding Statement and statements regarding the role of funders and authors contributions as follows:

Financial Disclosure Statement

The author(s) received no specific funding for this work. RP and SS are affiliated to a commercial company: BeSB GmbH. The funder provided support in the form of salaries for authors RP and SS but did not have any additional role in the study design, data collection and analysis, decision to publish, or preparation of the manuscript. The specific roles of these authors are articulated in the ‘author contributions’ section. 

Authors Contributions

Conceptualization: MT. Data curation: NS. Formal analysis: TBJ. Investigation: NS. Methodology: NS, TBJ, SS, MT. Project administration: NS. Resources: NS. Software: RP, SS. Supervision: MT. Validation: TBJ, MT. Writing - Original Draft Preparation: NS, TBJ. Writing - Review & Editing: NS, TBJ, SS, VV, MT.

Competing Interest 

TBJ and VV declare no competing interests. RP and SS are employed engineers at BeSB GmbH Berlin, and NS currently serves as unpaid consultant and informal scientific advisor for BeSB GmbH Berlin. The device presented and tested in this study is a product developed in collaboration between BeSB and MT with potential for commercialization. This commercial affiliation does not alter our adherence to PLOS ONE policies on sharing data and materials.

As suggested, we have also amended the methods section and ethics statement of the manuscript to state that the person depicted in Figure 4 provided consent for publication. The signed informed consent is submitted with the revised manuscript. 

Finally, we uploaded the minimal anonymized data set for each task administered in the study along with a description of all variables definitions as Supporting Information files. 

We believe that the adjustments implemented according to the feedback received have indeed increased the overall quality of the manuscript and we hope this revised version finds your approval.

We would be glad to respond to any further questions and comments that you and the reviewers may have and look forward to hearing from you regarding our submission.

Sincerely,

Dr. Nina Schaffert 

Response to Reviewer 1:

Overall comment: We greatly appreciate the constructive suggestions and have adjusted the manuscript accordingly. The revisions focused primarily on the introduction and discussion sections to clarify aspects relating to the motivation for the development of the SONATA and to better discuss points raised regarding the study results, future research and study limitations. We hope that these modifications have addressed your main suggestions and concerns. Please, see below point-by-point replies to your comments.

Reviewer #1: Data Availability: The form states, “Stating ‘data available on request from the author’ is not sufficient. If your data are only available upon request, select ‘No’ for the first question and explain your exceptional situation in the text box.” However, the authors stated, “The datasets generated and/or used during the current study are not publicly available but are available from the corresponding author on reasonable request.” No description of an exceptional situation was provided.

REPLY: The data sets are now available as supplementary material.

Major Comments:

- Throughout much of the manuscript, I found myself asking, “How is the SONATA different from existing music therapy devices? What is the key innovation?” The manuscript does a good job of motivating music-based rehabilitation but does not discuss the gaps that the SONATA addresses until the end of the manuscript. An earlier, structured review of relevant devices would be useful to motivate the development of the SONATA and clarify the message of the manuscript.

REPLY: Thank you for your suggestion. We added a new paragraph addressing some limitations of currently available devices and motivating the need for the development of the SONATA (Page 5). 

- Is there a reason the grid layout was chosen as opposed to a typical keyboard layout? Were other layouts tested, and, if so, did they have any impacts on the results? I also have similar questions regarding the size and shape of the virtual keys. (e.g., Can smaller keys be used to encourage greater spatial accuracy?)

REPLY: The device was conceptualized as a touch surface for motor training in therapy/rehabilitation, hence the spatial dimensions have to allow for mapping functional reaching movements of full elbow flexion/extension and shoulder adduction/abduction that are completed by touching a targeted sound button. After piloting in hemiparetic arm rehabilitation with persons with different body composition (size, gender), a 47x78cm surface seemed most appropriate to accommodate different body shapes. A typical keyboard layout would not accommodate those functional movements.

- Introduction, last paragraph: “With that in mind, a novel music-based therapeutic device for upper extremity movement training was developed to improve upper extremity motor function, to increase independent patient engagement, to enhance treatment quality, intensity, and compliance, and to assist therapists during treatment implementation and assessment.” This sentence suggests that the SONATA does these things, but they have not been validated and this study does not attempt to demonstrate them. The literature review suggests that these claims could be true for the SONATA, but the statement in its current form is misleading.

- Introduction, last paragraph: “to evaluate the feasibility of the system’s functioning for upper extremity movement training.” Given the subject set, this statement is also misleading.

REPLY: Thank you for drawing our attention to the word choices here. We have rephrased the objectives of the study to clarify any misleading points. The sentence now reads “Therefore, the objective of this study is to describe a novel music-based therapeutic device called SONATA, and to conduct a pre-clinical, single arm trial to evaluate the device’s functioning with healthy individuals” (Page 5).

- Methods, Section 2.3: “… thus assisting the therapist in assessing measurable changes in upper limb function throughout training.” This claim may need more justification. The experiment does not show that the data collected by the tasks, as they are implemented by the SONATA, correspond to changes in upper-limb function. How the therapists interact with this data (e.g., visualization, as opposed to raw data) is also important to determining if the data assists therapists. If it is too difficult to draw actionable inferences from the data, then in practice it will just be ignored.

REPLY: Thank you for your suggestion. Indeed, we cannot relate the data collected by the current study with changes in upper limb function. We modified the sentence with a more cautious wording. 

- Discussion, second paragraph: Is performance accuracy really an appropriate measure of ease of use? How hard does a device have to be to use to elicit a significant decrease in accuracy for young, healthy subjects? It also does not account for the interface design or therapist interactions with the system. Some sort of usability questionnaire or qualitative assessment may help to better support this claim. The limitations paragraph mentions the need for additional evaluation, but this statement is too strong.

REPLY: As suggested, we re-structured the discussion to clarify the main findings and better discuss aspects that need to be further addressed in future research. 

- Conclusions: “… indicate the feasibility and reliability of the device as a tool for upper extremity movement training.” This statement is too strong given the subject demographics and experimental design. More cautious language should be used.

REPLY: Thank you for drawing our attention to the word choices. We have completely rephrased the conclusions.

Minor Comments:

- Methods, Section 2.1: Please include a more thorough description of the hardware. For example, is the device battery-powered or plugged in? Is the system wirelessly controlled via Wi-Fi, Bluetooth, or some other standard? If the device was designed to minimize latency between user input and sound output, what was the average latency?

REPLY: We have added the missing information to the description of the devices, as suggested.

- Figure 1: It would be useful to readers to visually convey the scale of the devices. From the text, I see that the SONATA is rather large (approx. 0.7 m by 0.4 m), but it appears much smaller in this figure because of the adjacent tablet. (Though Figure 4 helps with this issue to some extent.) The figure should also more clearly identify which device is the SONATA and which is the tablet to prevent potential confusion.

REPLY: We have modified Figure 1 and added the dimensions of the devices accordingly and labelled the devices to avoid confusion.

- Figure 2: The authors may want to consider splitting or vertically arranging this figure so that the interfaces are easier to see. In comparison, Figure 3 was much easier to see.

REPLY: As suggested, we rearranged the figure and enlarged the panels to make them easier to see.

- Methods, line 160: “BPM” Please define abbreviations before use for clarity. Readers that are not familiar with music may not know that this means beats per minute.

REPLY: Corrected.

- Methods, Section 2.4.1, line 224: This is admittedly a bit of a nit-pick (i.e., the statement is acceptable as-is), but neurological damage or disorders can also affect upper limb movement.

REPLY: Thank you for this note. We have extended the sentence and included the information.

- Methods, Section 2.4.4: For clarity and readability, please avoid using variable names (e.g., “PAT_TIME”) without first defining them in English. At the minimum, a table of variable names and their descriptions could be provided and referenced before use. Avoiding the use of variable names altogether and using English instead is even more preferable.

REPLY: Thank you for the advice. As suggested, we have added a table with the variables names, their definition, abbreviation and the unit used as supplementary material. In addition, we removed the variable names and described the variables in plain language.

- Results, Task 4, lines 407–410: Is ms an appropriate unit here? Reporting seconds rounded to one significant digit after the decimal would be more easily parsed by the reader and would not change the conclusions.

REPLY: Corrected.

Spelling and Grammar: Below are a few miscellaneous items the authors may wish to address. This list is not necessarily comprehensive; the authors should do another proofreading pass.

- Introduction: “Post-stroke” is usually used as an adjective and not as a noun. The authors may want to consider saying “stroke” or “stroke survivors”.

REPLY: Corrected.

- Introduction, line 112: “With that in mind,” The “that” is ambiguous.

REPLY: We rephrased the sentence.

- Methods, line 130: “built-in”

REPLY: Corrected

- Methods, line 250: “pitches”

REPLY: Corrected

- Methods, line 282: “composed of six to nine”

REPLY: Corrected

- Methods, line 294: “composed of seven”

REPLY: Corrected

- Methods, line 295: “at a time”

REPLY: Corrected

- Decimal values should have a leading zero where appropriate. (e.g., “p < 0.05” on line 360.)

REPLY: Corrected

 

---------

Response to Reviewer 2:

Overall comment: We greatly appreciate the constructive suggestions and have adjusted the manuscript accordingly. The revisions focused primarily on the introduction and discussion sections to clarify aspects relating to the motivation for the development of the SONATA and to better discuss points raised regarding the study results, future research and study limitations. We hope that these modifications have addressed your main suggestions and concerns. Please, see below point-by-point replies to your comments.

Reviewer #2: You have described a tablet device designed for upper limb training and presented data collected from participants' hand and finger movements using the device who do not have upper limb impairment (normal controls).

Please add to the title 'a pre-clinical, single arm trial'. I would also consider revising the title further, to specify hand and finger or fine motor training, since there is no description in the manuscript of elbow or shoulder extension/flexion, aduction/abduction, or other gross motor movements.

REPLY: Thank you for your suggestion. We modified the title as suggested and adjusted the manuscript to specify that this study is a pre-clinical study that used tasks that require predominantly hand/finger movements. Nevertheless, we think that with the expanded task description and the added dimensions of the device, it will be clearer that the tasks do require gross movements as well.

In the abstract you use the terms neurologic recovery and neurologic conditions, please be more specific.

REPLY: We have adjusted the abstract as suggested. The term “neurologic recovery” now reads “neurological rehabilitation”, and we added examples of neurologic conditions where upper limb function can be compromised, such as cerebral palsy, stroke and Parkinson’s Disease.

Line 32, correct to 'built-in'

REPLY: Corrected.

Line 46, Abstract, this conclusion cannot be justified. please amend to reflect your findings from pre-clinical phase, i.e. it is not known whether this would work with patients who have UL paresis or apraxia.

REPLY: Thank you for the comment. We have rephrased the description of the results and conclusion of the abstract.

Line 55, just state 'Stroke'

REPLY: Corrected

Line 74: References 35, 37, 38, 39 are not TIMP as they do not require facilitating music for movement synchrony.

REPLY: Indeed, the highlighted references relate to studies using Music-supported Therapy and not TIMP. However, both techniques use musical instrument playing (although with different protocols) to train fine and gross movements of the paretic extremity. We clarified this point in the sentence.

Line 82. It is not clear what you are saying here. It seems that you are referring to other research here (Rojo et al, 2011, Ripolles et al, 2015, Altenmüller et al 2009, etc), where audio-motor coupling is discussed, which you cite later. Please revise the whole literature review to make it shorter and more succint, including only the key literature relevant to UL rehab (hand, finger, fine motor) and the use of rhythm-based interventions and technology such as keyboards and touchscreen instruments.

REPLY: Thank you for your suggestion. We excluded the highlighted sentence and attempted to focus the literature review on studies focusing on upper limb rehabilitation. 

Line 117. It needs to be completely clear what you have done throughout the manuscript, the this is pre-clinical testing, which can only show that the hardware and software is working with normal controls.

REPLY: Thank you for drawing our attention to the wording used here. We have rephrased the objectives of the study to clarify any misleading points. The sentence now reads “Therefore, the objective of this study is to describe a novel music-based therapeutic device called SONATA, and to conduct a pre-clinical, single arm trial to evaluate the device’s functioning with healthy individuals” (Page 5).

Line 130 correction 'built-in, and omit the word 'sound', just write 'speakers'

REPLY: Corrected.

139-142 graphically and acoustically program, please re-write to explain more clearly. Do you mean that the therapist (music therapist?) programs the screen layout, metronome setting and/or drawing exercise while the patient is perfomring the exercises? Please clarify.

REPLY: Yes, the therapist can program new patterns using the controller tablet while the client performs a different exercise in the SONATA touchscreen interface. We trust this clarifies the motivation for having two different interfaces.

161: do you mean synchronise their movements? Please clarify. What if they have cognitive (attention and exec function) impairment, and find the metronome distracting, performing UL movements better without it? This needs some discussion later 

REPLY: Yes, the metronome function allows movement synchronization and the feature can be disabled by the therapist according to the exercise objectives or the patient’s needs. A note to clarify that the metronome can be disabled was added to clarify this point (Page 9).

203 Please define what this (ASCII) file is. Readers really need to know that data collection and storage is secure and can conform to relevant policies to protect patient identity. Please clarify that your system is secure in this regard.

REPLY: Thank you for raising the point about data protection. We have included more information on the system’s security. Specifically, the recorded data is stored in the tablet’s password-secured internal memory and cannot be transmitted as the device is not connected to a network.

208 'The output files also contain timing and location information of each patient’s input.' clarify, does this mean time of the session and where it took place? If so, how are these data stored to ensure data protection?

REPLY: Thank you for this note. We have modified the sentence to clarify that only information on movement position and timing of keypresses are stored. Our word choice was not accurate in the previous version of the manuscript. 

210 'inter-stimulus interval, inter-response interval' please define

REPLY: A definition for both terms was included (Page 11)

219 'Functional tests were conducted with 21 healthy individuals (12 males, 8 females) recruited at...' this amounts to 20.

REPLY: Corrected.

228 No info here about reducing compensatory movements in people who actually have UL impairment, nor at any point later in the manuscript. This must be discussed to some degree.

REPLY: The subjects in the current study are healthy and not executing compensatory movements. With persons who will potentially use compensatory movements, 2 factors can be implemented to reduce/eliminate those movements:

- Research has shown that compensatory UE movements (trunk flexion, excessive shoulder abduction, circular arm trajectories) during reaching movements are reduced when movements are cued cyclically by auditory rhythm (Thaut et al, 2002 Neuropsychologia; Malcolm et al, 2009, Topics in Stroke Rehab). SONATA does not only produce auditory feedback at movement completion but also anticipatory rhythmic auditory movement cues. The perceptual-motor architecture of SONATA will probably provide compensatory movement reduction. This can be further tested in future research.

- Additionally, therapists or caregivers may manually assist by fixating trunk position.

268-269 youve cited MST rsearch but under the name of TIMP, this needs correcting, and here you need to reference MST to help identify what it is to the reader

REPLY: We have expanded the Introduction to make reference to both therapeutic techniques (MST and TIMP). The protocol description and references cited in this paragraph indeed refer to MST.

288-290 why such a cognitively taxing task when the focus is on UL rehab? How is this supported as a key component for UL rehab, any literature?

REPLY: Therapeutic protocols often implement tasks that vary in sequence length, movement velocity and direction, and these aspects commonly progressively increase in difficulty. For research purposes, here we combined all these features in a single task as an example of a motor training exercise that could be implemented with the SONATA. However, we acknowledge that this task requires preserved cognitive functions, thus task difficulty would need to be adjusted for patients with important cognitive function impairment. 

308 hypometria, bradykinesia, breif definition required

REPLY: A definition was added for both terms (Page 16).

309-312 this should be in the background section

REPLY: Thank you for your suggestion. We briefly described each task in the Introduction section.

328-329 (PAT_TIME – TICK_TIME) please define

334 SEQ_KEY and PAT_KEY please define

REPLY: We agree that including the names of the variables as found in the data output files can be confusing. We removed the variable names and described them in plain language. In addition, we have added a table with the variables names, their definition, abbreviation as supplementary material. 

376 'Nonetheless, the analysis indicated that there were significant main effects of

377 sequence length' This is useful data for informing on how to adapt the software/exercises for phase 1 clinical trial. There should be some discussion of this under 'future research' or 'clinical trial planning'

REPLY: Thank you for your suggestion. We included a new section in the Discussion to address these considerations regarding the study results and future research, as suggested.

379-381 Anyone working in neurorehab (occupational, physiotherapis for example) would anticipate this response and be able to set up exercises of appropriate complexity for patients, based on their assessment of cognitive function, before the session. Can you explain how this is adding to or enhancing current neurorehab provision, for example in stroke? 

REPLY: Our primary goal with this study was to describe the device and test whether the system’s hardware and software were functioning adequately. Therefore, in our view, the fact that our results corroborate the findings consistently reported in previous research is an indication that the data recorded by the device is reliable. 

382-384 Needs later discussion or some acknowledgement, as patients with cognitive impairment (attention and executive function) may not be able to synch with the pulse, depending on injury or stroke type and time post onset.

REPLY: We acknowledge that movement synchronization to a metronome may be difficult depending on the severity of the motor and/or cognitive impairment, type of injury or stage of the condition, and presented possible options that the therapist have to address this issue using the different functionalities of the device. 

387 Needs discussion in the context of how to adapt the equipment.software for clinical trials/use

REPLY: We included a new section in the Discussion to address important considerations about the task settings and future research, as suggested.

395 section on finger tapping task: Obviously this needs discussion in context of clinical trials, i.e. you need to test whether the data are accurate for those with apraxia for example

REPLY: Thank you for your point. We included this point regarding our study results in the discussion section.

419-421 please amend. feasibility and ease of use of the device are not demonstrated by participants’ performance accuracy. It is the fact that they could access and complete the exercises using this equipment (each clinical setting also needs to be taken into consideration, i.e. acute, subacute, home, carehome, etc) that demonstrates this, but only if you can describe the level of support that was required by the therapist/facilitator. Was lots of assistance required so that participants could follow the exercises? How has your trial informed on the next stage, clinical trial.

REPLY: Thank you for your comment. We amended the discussion regarding the results of the study to better reflect the findings of this pre-clinical study. 

422 'we replicated the results of previous research' what research? You state this previously, in the abstract. It is not clear.

REPLY: We clarified this sentence. In our humble opinion, the fact that the results of the functional tests concur with the findings consistently reported in previous research is an indication that the data recorded by the device is reliable.

440 is cognition consistently part of the focus throughout? Please ensure you maintain focus, what is primary, secondary, i.e.you need to state from the ofset that attention, executive function (memory you mention) are important to potential patient populations being able to access these exercises, that the clnician operating and delivering must consideer these functions and that there may be benefits in these domains from patients doing these exercises, as has been found in some MST research (Grau-sanchez for example).

REPLY: We have rephrased this entire paragraph to better discuss the points raised.

447-450 ' Studies have consistently demonstrated that active music playing is effective to train upper extremity movement [35-44] due to crucial elements, such as the display of real-time multisensory information [54-56] and the use of metronome or beat-enhanced music to support movement training [49,62,63], which promote neuroplasticity [39,57-61]...' The way this is written suggests that these citations (39, 57-61) involved metronome or beat synchronisation, which they didn’t so I suggest re-writing this.

REPLY: This sentence was removed from the revised version of the manuscript.

454-455 ...'including the availability of features that allow for better control and documentation of the training exercises implemented in each session.' No literature is cited here that indicates this is needed and this was not a clearly stated key aim at the ofset. What data supports a claim that this equipment achieves this? Have you clearly descibed how SONATA does this? Has it been reliability checked?

REPLY: We have completely rephrased this point.

458 but no correlation has been made between your data and standard assessment tools for UL function such as Fugl Meyer and ARAT? SO how do you know that this helps to achieve functional goals? PLease more clearly correlate, if possible, your data with that of MST studies where there is some correlation between tapping tasks and motor assessment outcomes.

REPLY: We have added the observation that our pre-clinical results were not correlated with other standard assessments as a limiting aspect of this study and a point that should be incorporated in future research.

458-459 this hasn’t been feasibiity tested though. You don't know how SONATA will perform in any setting. Please amend and state that this is a limitation of your study and/or that you need to conduct a clinical trial, most likely a feasibility study.

REPLY: Thank you for your comment. We amended the limitations to add the points suggested.

465 ...'transform the therapeutic process.' In the interest of not making exaggerated claims, in what context? One paper is cited hear, published in a music therapy journal. You have completed a pre-clinical testing study with normal controls, not patients. Please write in a way that does not imply that SONATA can achieve this transformation. So far, you can only use published data, primarily from MST studies.

REPLY: This sentence was removed.

472 Can you be clear, are you saying that studies are under way (recruiting) with clinical populations?

REPLY: We added a new section to discuss directions for future clinical research

475 need to be specific here and say what the limitations definitely were: our study was limited to testing the software and data collection. No data were correlated with standard UL measures, which needs to be done in a formal trial.

Exercises have not been tested for feasibility of delivery with stroke or other patient populations requiring neurorehabilitation in any clinical setting. Feasibility needs to be quantified in terms of acceptability of the exercises, tolerance and adherence. Following this, if data are favourable, a pilot or definitive trial would be required with a sufficiently powered sample.

Prior to that, since the device is intended for biomedical treatment, it would need to be registered as a medical device. The latter, must be acknowledged/discussed and is not mentioned at all throughout the manuscript, can you justify why it is not mentioned? If not, plese amend.

REPLY: Thank you for your constructive suggestion. We amended the section discussing the limitations of this study and included the points suggested.

481 '9 sequences' if you are presenting this as a potential tool to aid UL rehab, then you must discuss more about the implications. Since tolerance and adherence to treatment are central in enabling patients to achieve the required high dosage of treatment (MST = 3-4 weeks at 5 X per week) to show any clinical or statistical significance, wouldn’t 9 sequences per task fall rather short of achieving this?

REPLY: This paragraph was amended. Further improvements in the device hardware and software might be implemented as further user-experience evaluations are conducted.

487 'Upper extremity movement training' Please consider revising the manuscript to consistently state hand and finger and/or fine motor function. THink about where you want to take this research, what the next stage is. This study must support the next stage, if enough has been done here to pave the way, together with all pre-existing evidence.

REPLY: As suggested, we adjusted the manuscript to clarify the tasks implemented in the study.

488 'well known' be precise, clinically relevant. Are they well known? Where are the citations?

REPLY: This sentence was removed and rephrased to indicate that we adapted tasks that have been previously used in research and/or clinical practice and included the appropriate citations.

488 'test the feasibility, ease of use.' To a degree, but the data need to be collected alongside standard UL measures from people with UL impairment and reliability checked

REPLY: We rephrased our results to better reflect the findings of this pre-clinical study.

492 implementation and independent patient engagement ( I assume you mean self-delivery) these are two diferent things. Please clarify and define.

REPLY: This sentence was removed.

The conclusion could be better written. To say that results indicate feasibility and reliability is too general, too many areas are not covered in your study. What about positioning to mitigate compensatory movements, data protection (storage and transfer of data), checking of SONATA data alongside standard UL assessment tool data, infection control of equipment in clinical settings, who would deliver the exercises, a music therapist, any clinician, what can you conclude needs to be done next based on your findings, what dosage of SONATA exercises could be delivered in a clinical setting and are there currently enough exercises, how long does it take a clinician to program each exercises for each patient and what is the impact on managing a clincal case oad, what about medical device registration for SONATA?

REPLY: Thank you for your constructive feedback. We completely rewrote the conclusion section based on your suggestions.

The study might be a useful step in the right direction (adding clinical tools to help people with UL impairment to recover function), but as the manuscript stands it is not clearly framed and needs to be amended to consistenly adhere to the main areas of focus, purpose, findings, so that it is completely clear what the implcations are for future, related research, giving some indication of if and when SONATA might become a clinical tool.

REPLY: We thank the reviewer for your constructive comments and suggestions. We believe that the adjustments implemented according to your feedback have increased the overall quality of the manuscript and we hope this revised version finds your approval.

---

## [Editor Report · Decision Letter 1]

5 Nov 2020

Development and Evaluation of a Novel Music-Based Therapeutic Device for Upper Extremity Movement Training: A Pre-Clinical, Single-Arm Trial

PONE-D-20-23749R1

Dear Dr. Schaffert,

We’re pleased to inform you that your manuscript has been judged scientifically suitable for publication and will be formally accepted for publication once it meets all outstanding technical requirements.

Kind regards,

Sukru Torun

Academic Editor

PLOS ONE

Additional Editor Comments (optional):

Thank you for correcting and editing many important issues highlighted in our major revision suggestion and making the necessary adjustments to improve the scientific quality of your article.

Sincerely.
---

## [Editor Report · Acceptance letter]

9 Nov 2020

PONE-D-20-23749R1 

Development and evaluation of a novel music-based therapeutic device for upper extremity movement training: a pre-clinical, single-arm trial 

Dear Dr. Schaffert:

I'm pleased to inform you that your manuscript has been deemed suitable for publication in PLOS ONE. Congratulations! Your manuscript is now with our production department. 

Kind regards, 

on behalf of

Prof. Dr. Sukru Torun 

Academic Editor

PLOS ONE